# A Speech Enhancement Algorithm for Speech Reconstruction Based on Laser Speckle Images

**DOI:** 10.3390/s23010330

**Published:** 2022-12-28

**Authors:** Xueying Hao, Dali Zhu, Xianlan Wang, Long Yang, Hualin Zeng

**Affiliations:** 1Graduate Department, Wuhan Research Institute of Posts and Telecommunications, Wuhan 430074, China; 2Institute of Information Engineering, Chinese Academy of Sciences, Beijing 100093, China; 3School of Cyber Security, University of Chinese Academy of Sciences, Beijing 100049, China

**Keywords:** laser speckle image, speech enhancement, frequency spectrum correlation coefficient, frequency response

## Abstract

In the optical system for reconstructing speech signals based on laser speckle images, the resonance between the sound source and nearby objects leads to frequency response problem, which seriously affects the accuracy of reconstructed speech. In this paper, we propose a speech enhancement algorithm to reduce the frequency response. The results show that after using the speech enhancement algorithm, the frequency spectrum correlation coefficient between the reconstructed sinusoidal signal and the original sinusoidal signal is improved by up to 82.45%, and the real speech signal is improved by up to 56.40%. This proves that the speech enhancement algorithm is a valuable tool for solving the frequency response problem and improving the accuracy of reconstructed speech.

## 1. Introduction

In the field of information security and the prevention of social crimes, covertly acquiring clear remote speech signals has become an important research topic [1,2]. For years, researchers have proposed a variety of methods for measuring vibration signals [3,4,5,6,7,8,9,10,11], many of which are used in speech signal reconstruction and are based on laser. There are two main methods based on remote vibration signals detection: one is based on laser Doppler vibrometer (LDV) [3,4,5,6,7,8] and the other is optical sensing and image processing technologies [9,10,11]. LDV has the advantages of non-contact, high spatial and temporal resolution, real-time processing. But LDV cannot fulfill the full field detection. The optical sensing and image processing technology is mainly divided into two types: natural light image processing [12] and coherent light image processing [13,14,15,16]. The former has simple system structure and easy to implement. However, it requires high computational cost, and its processing speed is slow. In the 1980s, researchers of [13,14] proposed the method to extract subtle motion from speckle images. Additionally, in recent years, researchers have also reconstructed speech based on speckle images [15,16]. The principle of this method is to irradiate the rough surface object near the sound source with laser, and the reflected light interferes to form secondary speckle [17]. Then the sound makes the surrounding objects vibrate slightly, which makes the speckle pattern shift slightly. The remote speech signal is reconstructed by extracting the small movement between speckle images. The system of remote speech reconstruction based on laser speckle pattern is mainly divided into two parts: the first is the construction of an optical system, which is used to collect continuous speckle image sequences; the second is the design of a reconstruction algorithm, which is used to detect movement from speckle images and reconstruct remote speech signal.

As regards optical system construction, ref. [18] proposed a set of simplified optical equipment, it reduces the cost and realizes full-field non-contact detection. This optical device makes the technology more suitable for remote monitoring. The current laser technology and imaging technology can collect high-quality speckle images at a low price.

Regarding speech reconstruction algorithm, researchers have proposed a variety of methods, including digital image correlation (DIC) [19,20,21,22,23,24], optical flow method [16,25,26], and intensity method [27,28]. Ref. [29] proposed a geometric method to explain the motion of speckle. Ncorr [20] is a digital correlation algorithm specially designed by researchers for two-dimensional images. In recent years, with the rapid development of machine learning and artificial intelligence technology, many neural-network-based methods have been proposed, such as convolutional neural network (CNN) [30,31,32] and convolutional long short-term memory (LSTM) [33].

At present, the detection method based on laser speckle images is the most suitable for this task, but the problem of frequency response has been found in the actual experiments and applications. We found that the speech signal reconstructed by this optical device [18] has inhomogeneous enhancement or attenuation of speech components at different frequencies, and the frequency response of different vibration objects is different. Any object has a natural frequency, when the vibration sound wave of this natural frequency is transmitted to the object, the vibration amplitude of the object will have the maximum growth. Different objects have different natural frequencies, so the performance of frequency response is also different. This phenomenon greatly affects the accuracy of the reconstructed sound signal.

At present, many reconstruction algorithms ignore the frequency response, fewer algorithm is proposed to solve this problem. In this paper, a speech enhancement algorithm for different vibration objects is proposed to weaken the frequency response and improve the accuracy of speech reconstruction.

This paper has two main contributions.
The frequency response of long-distance speech reconstruction based on laser speckle image is proposed.Our algorithm is a speech enhancement algorithm designed to reduce the influence of frequency response, which greatly improves the accuracy of reconstructed speech signals.

The rest of this paper is organized as follows: Section 2 introduces the methodology of this paper, including DIC method and speech enhancement algorithm. Section 3 present the experimental setup. Section 4 introduces the experimental datasets and evaluation metrics. Section 5 is the result. Section 6 is the conclusion.

## 2. Methodology

### 2.1. Digital Image Correlation Method

There is room for improvement in many speech reconstruction algorithms. For example, the intensity method runs fast but has low accuracy. The performance of the six algorithms is compared in [34], and the results show that the cross-correlation method is one of the best choices. In order to accurately detect the frequency response in this optical device, this paper uses the DIC method as the benchmark.

Figure 1 shows the conversion between object vibration and speckle pattern displacement. In this figure, the transversal plane is regarded as the xoy plane, and the axial axis is the z-axis. The motion of this object can be divided into three directions: transverse, axial, and tilt. Ref. [18] proved that when strongly defocus the speckle image captured by the camera, only the tilt motion has the noticeable impact on the displacement of the speckle image, the influence of the other two motions on the shape and displacement of the speckle image can be ignored.

Refs. [35,36] deduced the speckle formation theory in detail, and ref [18] mentioned that making the imaging plane move from $Z_{1}$ to $Z_{2}$. $A\left(x_{o},y_{o}\right)\;$is the amplitude distribution of speckle at $Z_{2}$.
(1)$$A\left(x_{o},y_{o}\right)=\left|\iint exp\left[i\phi \left(x,y\right)\right]exp\left[i\left(\beta_{x}x+\beta_{y}y\right)\right]exp\left[\frac{-2\pi i}{\lambda Z_{2}}\left(xx_{o}+yy_{o}\right)\right]dxdy\right|$$
(2)$$\beta_{x}=\frac{4\pi tan\;\alpha_{x}}{\lambda }$$
(3)$$\beta_{y}=\frac{4\pi tan\;\alpha_{y}}{\lambda }$$
where λ is the optical wavelength and ϕ is the random phase generated by the rough surface. $\alpha_{x}$, $\alpha_{y}$ are tilting angle in the x-axis and y-axis respectively.

The inverse of the magnification of the imaging system is expressed in M.
(4)$$M=\frac{Z_{3}-F}{F}\approx \frac{Z_{3}}{F}$$
where F is the focal length of the imaging lens.

The speckles pattern displacement d is related to the tilting angle α of the object.
(5)$$d=\frac{Z_{2}\alpha }{M}$$

According to the previous description, we know that under appropriate conditions, there is only speckle movement between two consecutive speckle images, but no shape change. And the displacement is in the xoy plane. The relationship between two consecutive speckle images $I_{t},\;I_{t+1}$ is:(6)$$I_{t+1}=I_{t}\left(x+∆x,\;y+∆y\right)$$
where ∆x, ∆y are the relative displacements in the x-axis and the y-axis direction.

The calculation steps of DIC method are shown in Figure 2**.** Take two consecutive speckle images $I_{t}$ and $I_{t+1}$. Calculate the correlation coefficient between two images. At the maximum correlation, the corresponding displacement is the relative displacement between two images.

Corr2 is a function of finding the correlation coefficient between two matrices. The correlation coefficient r can be expressed as:(7)$$r(A,B)=corr2(A,B)=\frac{\sum_{m}\sum_{n}\left(A_{mn}-\bar{A}\right)\left(B_{mn}-\bar{B}\right)}{\sqrt{\left(\sum_{m}\sum_{n}{\left(A_{mn}-\bar{A}\right)}^{2}\right)\left(\sum_{m}\sum_{n}{\left(B_{mn}-\bar{B}\right)}^{2}\right)}}$$
where A and B are two-dimensional matrices of the same size, $\bar{A}$ is the mean of the matrix A, and $\bar{B}$ is the mean of the matrix B.

### 2.2. Speech Enhancement Algorithm

We use a sinusoidal signal which frequency varies from 80 to 1600 Hz and amplitude remains constant as the audio at the sound source (Figure 3 shows the waveform and frequency spectrum of the sinusoidal signal), and collect the corresponding speckle image dataset. Then use the algorithm in Section 2.1 to reconstruct the sinusoidal signal when the vibration object is a carton (Figure 4 shows the waveform and frequency spectrum of the reconstructed signal). By comparing Figure 3 with Figure 4, we found that the amplitude of the reconstructed signal is different at different frequencies, which seriously affects the accuracy of the reconstructed speech signal. Figure 5 is the reconstructed signal when the vibration object is a paper cup. By comparing Figure 4 with Figure 5, we found that the frequency responses of two different vibration objects are also different.

According to this phenomenon, we calculated the amplitude changes of several common vibration objects at different frequencies. According to the degree of change, the reconstructed speech signals of different frequencies are enhanced to different degrees. The above method can reduce the influence of the frequency response. The design flow of the algorithm is shown in Figure 6. To more clearly explain the steps involved in this speech enhancement algorithm, Figure 7 shows the results of each step of the speech enhancement algorithm when the vibration object is a carton.
Sinusoidal signal: Table 1 shows the common frequency range of speech. The sinusoidal signal which frequency varies from 80 to 1600 Hz and amplitude remains constant at 1.Capture laser speckle images: Use the high-speed camera to collect the corresponding laser speckle images.DIC method: Speech signals are reconstructed from speckle images using the DIC method described in Section 2.1. The discrete frequency domain sequence of the sinusoidal signal reconstructed is $s\left(f\right).$Connect local highest point: Connect the local highest points of the amplitude signal in the reconstructed speech spectrum to obtain its envelope signal. The envelope signal is named $E\left(f\right).$Enhancement signal: Equation (8) shows the calculation formula of the signal $h\left(f\right)$. Note that $E\left(f\right)$ at some frequencies is very small, which will make $h\left(f\right)$ very large. When $h\left(f\right)$ is greater than 1000, set it to 1.
(8)$$h\left(f\right)=\frac{E{\left(f\right)}_{max}}{E\left(f\right)}$$Real speech domain: Multiply the discrete frequency domain sequence of the reconstructed real speech $r_{1}\left(f\right)$ with the signal $h\left(f\right)$. The enhanced speech discrete frequency domain sequence is $R\left(f\right)$. According to the frequency range of speech, we process the speech with the frequency of 80–1400Hz.
(9)$$R\left(f\right)=r_{1}\left(f\right)\times h\left(f\right)$$

After the following steps, the discrete frequency domain sequence $R\left(f\right)\;$of the real speech signal will be obtained, and the speech waveform after speech enhancement can be obtained by using the inverse discrete Fourier transform (Equation (10)):(10)$$r_{n}=\frac{1}{N}\sum_{k=1}^{N}R_{k}e^{2\pi i\left(n-1\right)\left(k-1\right)/N}$$
where N is the total number of samples of the R sequence, R is the speech discrete frequency domain sequence after speech enhancement, and r is the speech discrete time domain sequences after speech enhancement.

## 3. Experimental Setup

According to previous research results, the formation and acquisition process of laser speckle image can be set (as Figure 8). Speckle is a three-dimensional ellipsoidal shape with its long axes facing the light propagation direction. Figure 9 shows the experimental platform. Figure 10 and Figure 11 are the equipment simulation diagrams of two kinds of lasers, and Figure 12 is the equipment physical map. The equipment used mainly includes:The high-speed camera MVCAM AI-030U815M, with a maximum frame rate of 3200 frames per second (fps);He-Ne laser (detailed parameters are shown in Table 2);Fiber laser (detailed parameters are shown in Table 3);Machine vision experiment frame, with fine-tuning camera clip and universal clip;One personal computer (PC) with universal serial bus 3.0 (USB3.0) interface.

In this experiment, the high-speed camera is used to collect speckle images. The frame rate of the camera is closely related to the exposure interval. The process of speckle image acquisition by high-speed charge coupled device (CCD) camera can be regarded as uniform sampling of continuous speckle video. The sampling frequency $f_{s}$ of the high-speed camera and the highest frequency of speech $f_{m\;}$ need to satisfy the Nyquist theorem (Equation (11)).
(11)$$f_{s}\geq 2f_{m}$$

Equation (11) shows that the frame rate of the high-speed camera should be greater than or equal to two times of the highest frequency of speech. The data in Table 1 show the speech frequency range of male and female, and $f_{m}$ = 1200 Hz. This paper considers that all the frequency ranges in the speech can be restored to meet the actual use requirements, so the frame rate of the high-speed camera is greater than or equal to 2400 fps. The frame rate of the high-speed camera used in this experiment is 3200 fps, which meets the basic requirements of speech reconstruction.

In conjunction with Figure 1, we describe various physical parameters of speckle images. The first is the resolution of the speckle patterns in the $Z_{2}$ plane:(12)$$\delta_{x}=\frac{\lambda Z_{2}}{D}\cdot \frac{1}{M}=\frac{\lambda F}{D}\cdot \frac{Z_{2}}{Z_{3}}$$
where D is the diameter of the laser beam, $Z_{2},\;Z_{3}$ are different distances in Figure 1. F is the focal length of the lens. λ is the optical wavelength.

The optical system has requirements for the focal length F. The size of the pixel in the detector is Δs. It is assumed that every speckle in this plane will be observed at least by K pixels. The requirements for F are as follows:(13)$$F=\frac{K∆sZ_{3}D}{Z_{2}\lambda }$$

The distance $Z_{2}$ needs to be satisfied:(14)$$Z_{2}>\frac{D^{2}}{4\lambda }$$

Finally, the number of speckles in every dimension of the spot is N:(15)$$N=\frac{\phi }{M\delta_{x}}=\frac{\phi D}{\lambda Z_{2}}=\frac{F\cdot D}{F_{\#}\lambda Z_{2}}$$
where ϕ is the is the diameter of the aperture of the lens, $F_{\#}$ is the F number of the lens, $M\delta_{x}$ represents the speckle size obtained on the $Z_{2}$-plane.

## 4. Experiment Datasets and Evaluation Metrics

### 4.1. Data Collection

#### 4.1.1. Sinusoidal Datasets

According to the speech frequency range in Table 1, we set the frequency of the sinusoidal signal to gradually change from 80 to 1600 Hz, with an amplitude of 1. Irradiate the laser generated by the He-Ne laser on five vibration objects near the sound source: carton, A4 paper, plastic cup, paper cap, and leaf. Use the CCD camera to collect the speckle image sequence generated. The number of speckle images is shown in Table 4.

#### 4.1.2. Two Laser Datasets

In order to verify the versatility of the algorithm under different laser systems, we use the He-Ne laser and the fiber laser to irradiate the carton near the sound source, and the number of speckle images collected is shown in Table 5.

#### 4.1.3. Five Vibration Object Datasets

To verify the performance of the algorithm under different vibration objects, we use real speech as the sound source signal, project the laser onto different vibration objects, and collect the corresponding laser speckle image sequences respectively. The number of speckle images collected by five vibration objects is shown in Table 6.

### 4.2. Evaluation Metrics

This paper needs to calculate the correlation between the original audio signal and the speech signal reconstructed from speckle images. Since the start time of the two signals cannot be accurately aligned, the waveform correlation coefficient in the time domain is meaningless. To evaluate the accuracy of a reconstructed speech, we use the frequency spectrum correlation coefficient as the evaluation metrics. The frequency spectrum correlation coefficient here refers to the amplitude correlation coefficient of the frequency spectrum in detail.

As we know, scaling speech amplitude will affect the loudness of the speech, not the timbre. Therefore, using the frequency spectrum correlation coefficient can quantify the accuracy of speech reconstruction.

The value of the correlation coefficient is between -1 and 1. When it is 1, the two speeches are equal or proportional. When it is −1, one speech is equal to or proportional linearly to another negative speech.
(16)$$\rho \left(x,y\right)=\frac{1}{N-1}\sum_{i=1}^{N}\left(\frac{x_{i}-\mu_{x}}{\sigma_{x}}\right)\left(\frac{y_{i}-\mu_{y}}{\sigma_{y}}\right)$$
where x and y represent the amplitude signal of discrete frequency spectrum signals with the same sampling points. N is the total number of samples. $\mu_{x}$, $\mu_{y}$ is the mean value of x and y accordingly. $\sigma_{x}$, $\sigma_{y}$ is the standard deviation of x and y accordingly. ρ is the correlation coefficient.

## 5. Results

### 5.1. Performance on Sinusoidal Datasets

We recorded the changes of the frequency spectrum correlation coefficient between the reconstructed speech and the original speech after using the speech enhancement algorithm. We used the speech enhancement algorithm in this paper to process the sinusoidal datasets to verify whether the algorithm can increase the frequency spectrum correlation coefficient between the reconstructed sinusoidal signal and the original sinusoidal audio. Table 7 shows the sinusoidal signal reconstruction accuracy in detail. Figure 13 shows the line graph of the frequency spectrum correlation coefficient of reconstructed sinusoidal signal on different vibration objects. The existence of frequency response has a great impact on the reconstruction of sinusoidal signal, which will make the spectral correlation coefficient low. The sinusoidal signal dataset only verifies whether our proposed speech enhancement algorithm can reduce the frequency response; it cannot be verified whether the algorithm can also play a positive role in real speech. Therefore, the following experimental results are used to verify the enhancement effect of this algorithm on real speech.

### 5.2. Performance on Two Laser Datasets

Using the same speech as the as the audio at the sound source, two lasers are used to irradiate the same vibration object (carton), and the CCD camera is used to collect the laser speckle image sequences. Table 8 is the comparison of the frequency spectrum correlation coefficient with and without speech enhancement. Figure 14 shows the reconstructed speech and original speech waveforms using the He-Ne laser. Figure 15 shows the reconstructed speech and original speech waveforms using the fiber laser. In the waveform comparison, we can also clearly find that the waveform of the speech is significantly improved after the speech enhancement.

### 5.3. Performance on Five Vibration Object Datasets

Table 9 shows the difference in frequency spectrum correlation coefficient between the reconstructed speech signal and the original speech using DIC and DIC + speech enhancement. The results show that after using the speech enhancement algorithm, the accuracy of speech signals reconstructed from different vibration objects is improved. This further proves that our algorithm has strong adaptability to different environments and has the practical application value.

The results in Table 9 show that the accuracy of speech reconstruction for some datasets is improved slightly after using speech enhancement algorithm. We analyzed the reasons for this phenomenon as follows: when collecting these datasets, the surrounding environment has little interference, which ultimately makes the accuracy of speech reconstruction high. This can also be proved by the fact that the frequency spectrum correlation coefficient before speech enhancement is higher than that of other datasets. Figure 16 shows the line graph of the frequency spectrum correlation coefficient of reconstructed speech signals on different vibration objects.

### 5.4. Discussions

Our speech enhancement algorithm is the speech process after the speech reconstruction algorithm, so whether the algorithm can be used in real time depends on the speed of the speech reconstruction algorithm. In the following content, we discuss some key factors that affect the performance of speech enhancement algorithm:(1)Loudspeaker performance: In order to provide a benchmark for frequency response, we use a sinusoidal signal which frequency varies from 80 to 1600 Hz and amplitude remains constant at 1 as the audio at the sound source. However, there will be errors in the sinusoidal signal played by the loudspeaker. These errors will be fewer if the loudspeaker performance is better.(2)Environmental noise and platform vibration: In the experiment, we find that the environmental noise and the vibration of the experimental platform seriously interfere with the sinusoidal signal reconstruction, and then affect the frequency spectrum correlation coefficient between the original sinusoidal signal and the reconstructed signal. The reconstructed sinusoidal signal contains environmental noise and vibration of the experimental platform, but the original signal is a clean and impurity-free speech, and the correlation coefficient between the two is low. Therefore, a quiet environment and a stable experimental platform are conducive to the verification of experimental results.

## 6. Conclusions

The main innovation of this paper is to propose a speech enhancement algorithm to solve the frequency response problem of speech signal reconstruction from laser speckle images. The experimental results in this paper show that our speech enhancement algorithm can effectively improve the accuracy of reconstructed speech, and the comparison of the reconstruction accuracy of different vibration object datasets verifies the practicability of the algorithm. In the experimental results, we found that when the ambient noise is small and the vibration interference of the experimental platform is small, the reconstructed speech accuracy is high. Our algorithm achieves a major advance of reconstructing speech from laser speckle images.

Although our speech enhancement algorithm has an obvious effect on reducing the frequency response problem, it also introduces some noise signals while increasing the accuracy of the reconstructed speech. In order to obtain higher-definition speech signals, we also need to make further improvements to the speech enhancement algorithm. In the future, we will focus on reconstructing speech with a small frequency response and high definition.

## Figures and Tables

**Figure 1 sensors-23-00330-f001:**
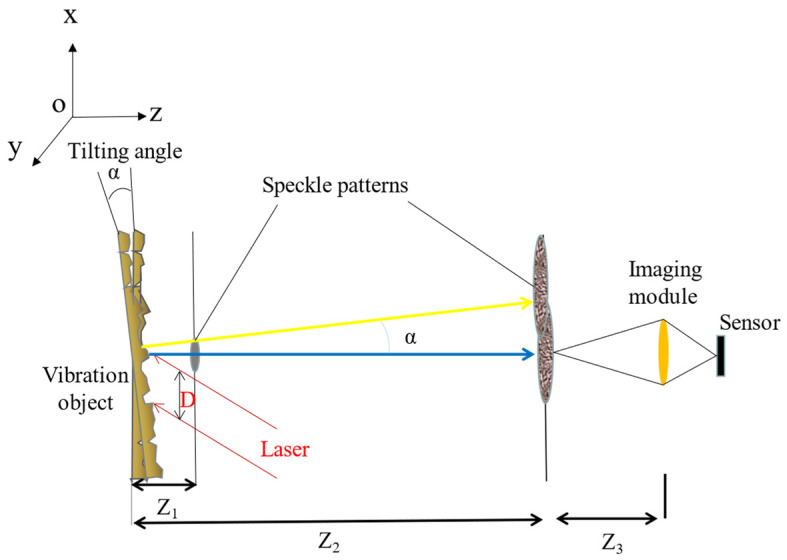
Theoretical description of the system. The conversion between object vibration and speckle pattern displacement.

**Figure 2 sensors-23-00330-f002:**
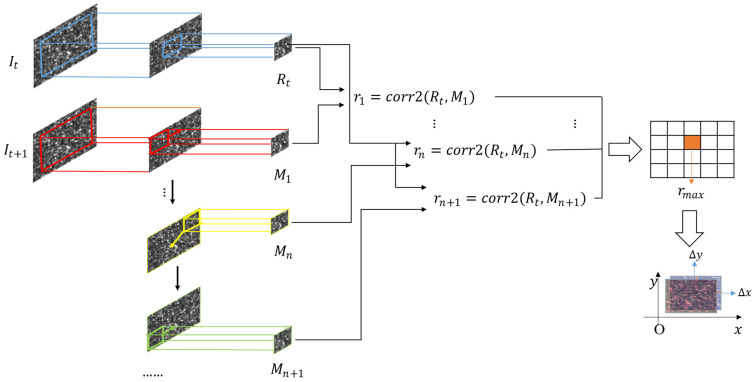
Digital image correlation method.

**Figure 3 sensors-23-00330-f003:**
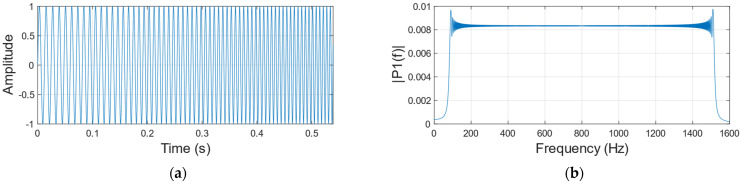
Original sinusoidal signal. (**a**) Time domain diagram. (**b**) Frequency domain diagram.

**Figure 4 sensors-23-00330-f004:**
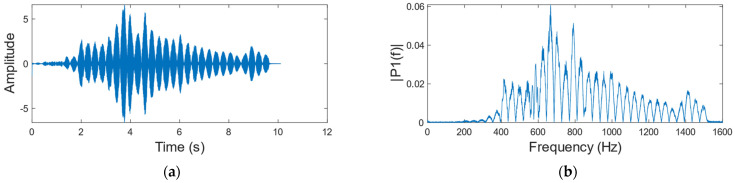
Reconstructed sinusoidal signal when a carton is the vibration object. (**a**) Time domain diagram. (**b**) Frequency domain diagram.

**Figure 5 sensors-23-00330-f005:**
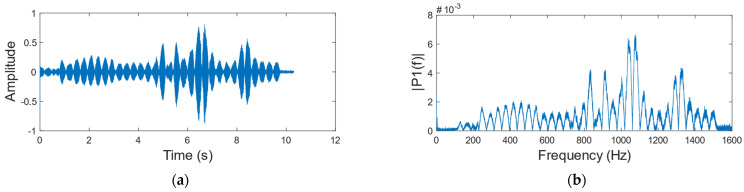
Reconstructed sinusoidal signal when a paper cup is the vibration object. (**a**) Time domain diagram. (**b**) Frequency domain diagram.

**Figure 6 sensors-23-00330-f006:**
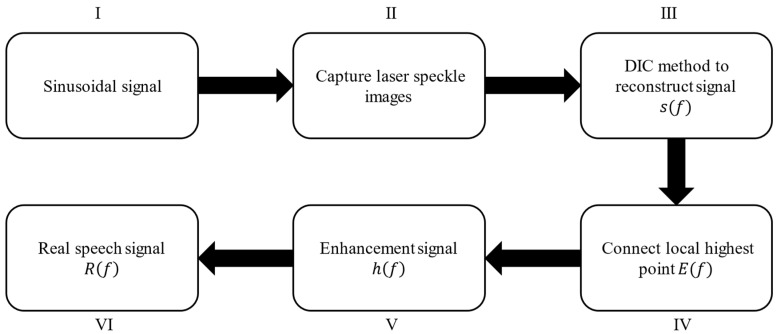
Speech enhancement algorithm.

**Figure 7 sensors-23-00330-f007:**
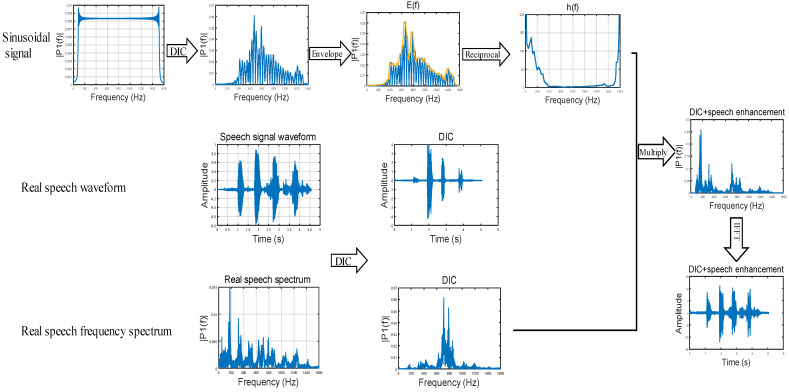
The process of speech enhancement when the vibration object is a carton.

**Figure 8 sensors-23-00330-f008:**
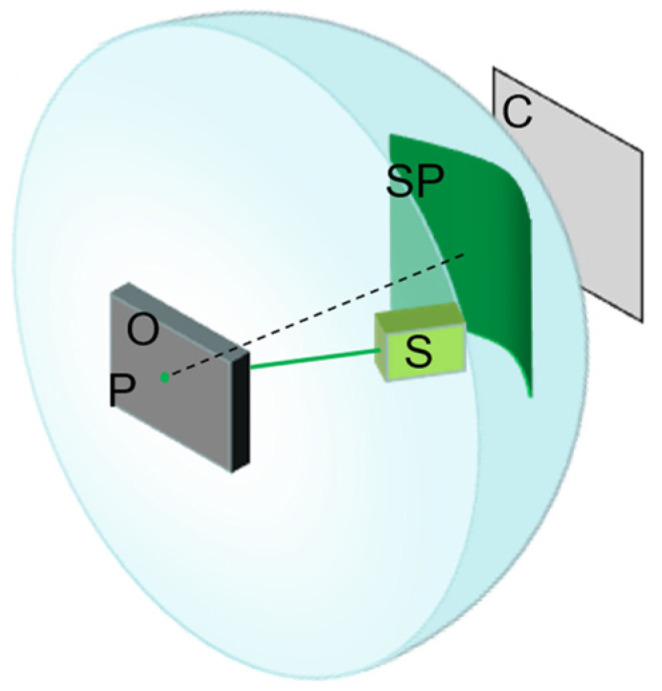
Schematic diagram of speckle image formation and collection. The laser beam generated by laser S irradiates on object O with a rough surface near the sound source (P is a point on object O surface), and then the reflected light interferes to form speckle. C is the camera sensor (the size is exaggerated).

**Figure 9 sensors-23-00330-f009:**
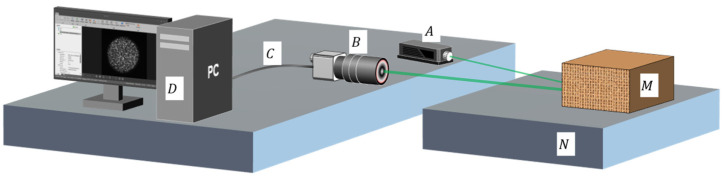
Experimental setup diagram. The laser beam generated by laser A is projected onto object M on the N platform. The change of speckle is collected by B camera, and these images are transmitted to the D computer through the C interface to extract the movement between speckle images.

**Figure 10 sensors-23-00330-f010:**
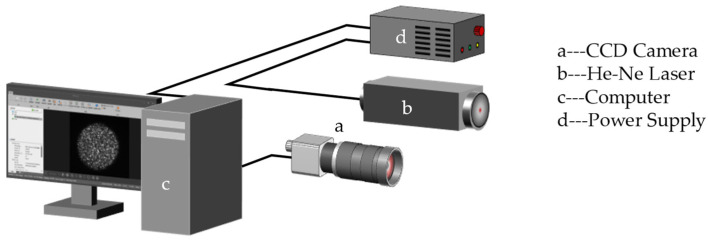
Simulation diagram of the He-Ne laser system.

**Figure 11 sensors-23-00330-f011:**
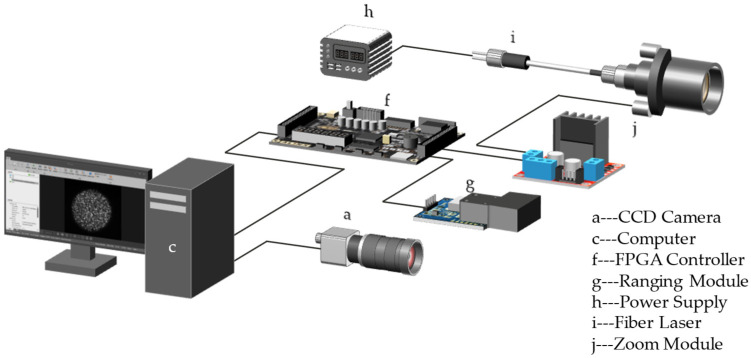
Simulation diagram of the fiber laser system.

**Figure 12 sensors-23-00330-f012:**
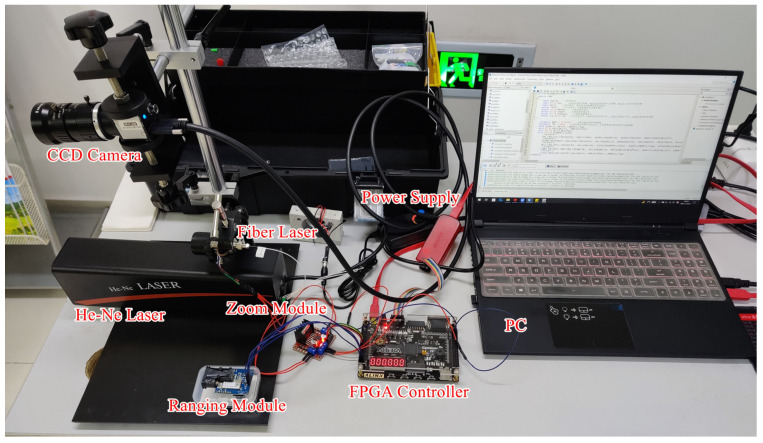
Equipment physical map.

**Figure 13 sensors-23-00330-f013:**
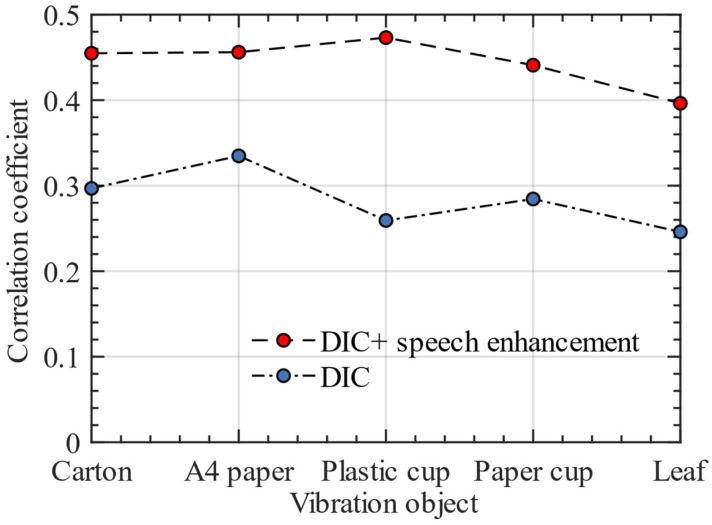
Accuracy comparison of sinusoidal signal reconstruction with and without enhancement.

**Figure 14 sensors-23-00330-f014:**
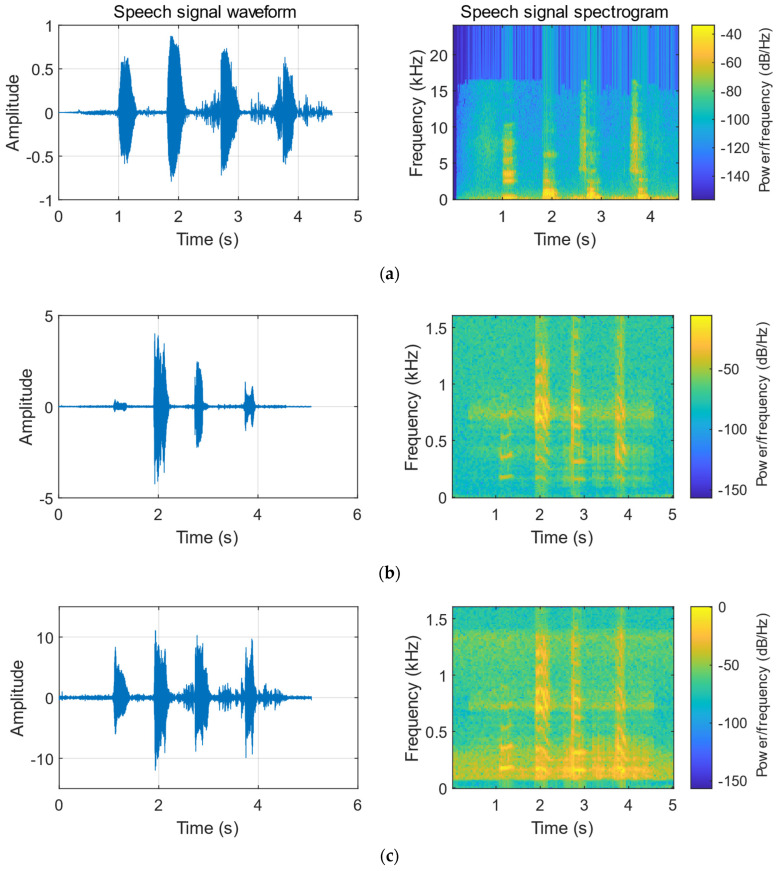
Comparison of speech signals reconstructed using the He-Ne Laser. (**a**) Original speech signal. (**b**) Speech signal reconstructed by DIC. (**c**) Speech signal reconstructed by DIC + speech enhancement.

**Figure 15 sensors-23-00330-f015:**
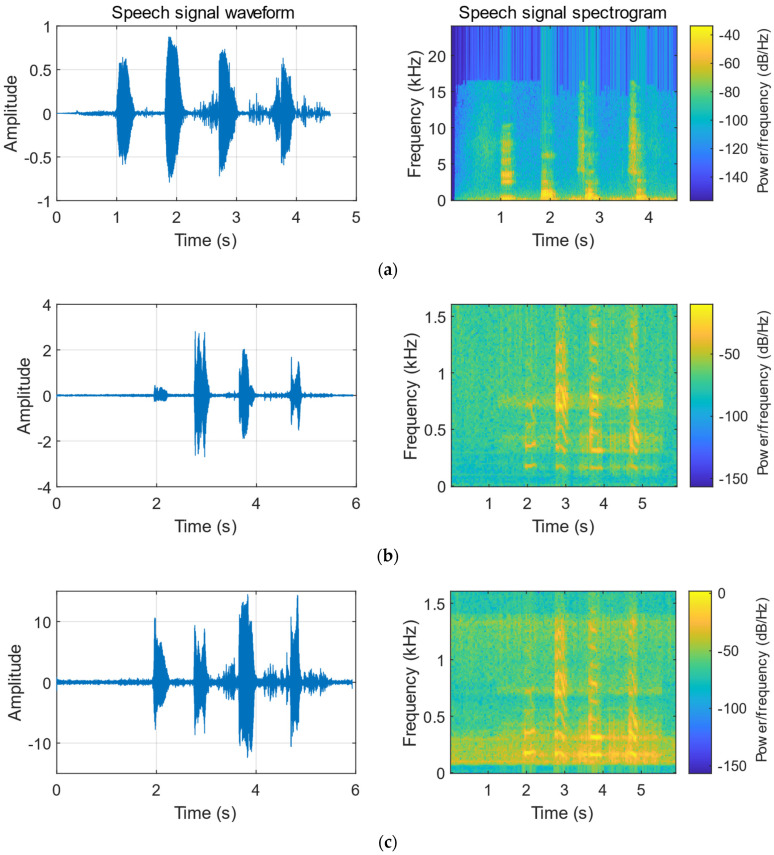
Comparison of speech signals reconstructed using the fiber laser. (**a**) Original speech signal. (**b**) Speech signal reconstructed by DIC. (**c**) Speech signal reconstructed by DIC + speech enhancement.

**Figure 16 sensors-23-00330-f016:**
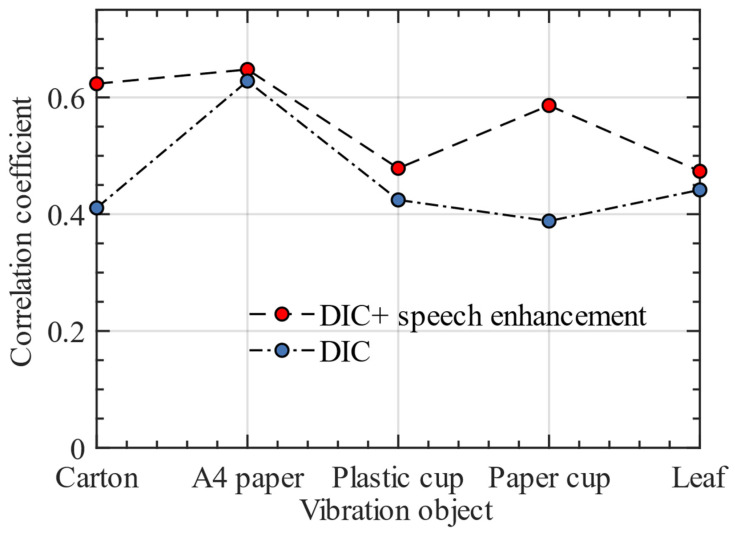
Accuracy comparison of speech signal reconstruction with and without enhancement.

**Table 1 sensors-23-00330-t001:** Frequency range of common speech (Hz).

Sound Area	Male	Female
Bass	82–392	82–392
Midrange	123–493	123–493
Treble	164–698	220–1100
Fundamental frequency range	64–523	160–1200

**Table 2 sensors-23-00330-t002:** Basic parameters of the He-Ne laser.

Parameter	Quantity	Unit
Wavelength	632.8	nm
Operating current	4~6	mA
Rated voltage	220 ± 22	V
Rated frequency	50	Hz
Rated input power	<20	W

**Table 3 sensors-23-00330-t003:** Basic parameters of the fiber laser.

Parameter	Quantity	Unit
Center wavelength	635	nm
Continuous output power	60	mW
Operating voltage	2.55	V
Threshold current	70	mA
Operating current	190	mA

**Table 4 sensors-23-00330-t004:** The number of speckle images acquisitions for sinusoidal datasets.

Vibration Object	Number of Speckle Images	Acquisition Time (s)
Carton	32,344	10.1
A4 paper	35,952	11.2
Plastic cup	31,941	10.0
Paper cup	32,970	10.3
Leaf	33,427	10.4

**Table 5 sensors-23-00330-t005:** The number of speckle images acquisitions for two laser datasets.

Laser	Number of Speckle Images	Acquisition Time (s)
He-Ne laser	16,213	5.1
Fiber laser	18,975	5.9

**Table 6 sensors-23-00330-t006:** The number of speckle images acquisitions for five vibration object datasets.

Vibration Object	Number of Speckle Images	Acquisition Time (s)
Carton	18,140	5.7
A4 paper	16,785	5.2
Plastic cup	19,443	6.1
Paper cup	19,247	6.0
Leaf	18,647	5.8

**Table 7 sensors-23-00330-t007:** Comparison of sinusoidal signal reconstruction accuracy.

Vibration Object	DIC	DIC + Speech Enhancement	Improve
Carton	0.2968	0.4548	53.23%
A4 paper	0.3348	0.4561	36.23%
Plastic cup	0.2593	0.4731	82.45%
Paper cup	0.2844	0.4409	55.03%
Leaf	0.2460	0.3963	61.10%

**Table 8 sensors-23-00330-t008:** Comparison of the speech reconstruction accuracies of two laser datasets.

Laser	DIC	DIC + Speech Enhancement	Improve
He-Ne laser	0.3624	0.5668	56.40%
Fiber laser	0.4155	0.5849	40.77%

**Table 9 sensors-23-00330-t009:** Comparison of the speech reconstruction accuracies of five vibration object datasets.

Vibration Object	DIC	DIC + Speech Enhancement	Improve
Carton	0.4107	0.6234	51.79%
A4 paper	0.6280	0.6478	3.15%
Plastic cup	0.4244	0.4786	12.77%
Paper cup	0.3883	0.5862	50.97%
Leaf	0.4416	0.4735	7.22%

## Data Availability

Not applicable.

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
