# Peer review of "A Speech Enhancement Algorithm for Speech Reconstruction Based on Laser Speckle Images"

_sensors, 2022, doi:10.3390/s23010330_

Round 1

Reviewer 1 Report

This paper introduces a method using laser speckle analysis and a new speech enhancement algorithm to reconstruct speech signals with high accuracy. Overall, this is an interesting paper with sufficient scientific value. Some questions need to be further explained or addressed.

1.      Regarding the experimental setup, please add information of these parameters: imaging magnification, minimum speckle size (refer to ‘Laser speckle contrast imaging in biomedical optics’, JBO, 2010, Equation 10 for calculation), exposure time.

2.      The explanation and math description of speckle pattern need to be further improved. The authors can refer to ‘Journal of Biomedical Optics, 2020, 25(5): 055003’, equation (3) and (4). Both references above should be properly cited.

3.      How is the number of speckles calculated in Table 4, 5, 6? Is it the number of images of speckle pattern or the number of small speckles of each image?

4.      Is the speech reconstruction algorithm applied after all data was acquired or is it a real-time algorithm that can be applied during acquisition?

5.      A discussion section is missing. There are quite a lot of relevant discussion can be presented. For example, any disadvantages of the proposed methods and how they can be overcome, how laser speckle size can affect the accuracy, how environment noise (wind, mechanical vibration etc.) affects the reading.

6.      The speech similarity ranges from 0.2 to 0.6 in the paper. Are the values comparable to the industry standard or main stream academic standard? A more detailed comparison / benchmarking of the performance should be added.

Reviewer 2 Report

In this papers authors claim that nowadays, all kinds of speech reconstruction algorithms ignore the frequency response, there not existing a proposed algorithm to solve this problem. To overcome this problem, the authors propose a speech enhancement algorithm, based on the speckle analysis, to reduce the frequency response caused by the detection systems. They claim that after using their algorithm, the spectral similarity of the reconstructed sinusoidal signal is improved by up to 92.49%, and the real speech signal is improved by up to 58.84%. This approach is novel and could be published in the Sensor Journal, after constructive criticisms to improve your contribution.
